# Combined translational and rotational perturbations of standing balance reveal contributions of reduced reciprocal inhibition to balance impairments in children with cerebral palsy

**Jente Willaert**[1]*, **Kaat Desloovere**[2], **Anja Van Campenhout**[3], **Lena H. Ting**[4,5], **Friedl De Groote**[1]

**1** Department of Movement Sciences, KU Leuven, Leuven, Belgium, **2** Department of Rehabilitation Sciences, KU Leuven–UZ Leuven, Leuven, Belgium, **3** Department of Development and Regeneration, KU Leuven–UZ Leuven, Leuven, Belgium, **4** Wallace H. Coulter Department of Biomedical Engineering, Emory University and Georgia Tech, Atlanta, Georgia, United States of America, **5** Division of Physical Therapy, Department of Rehabilitation Medicine, Emory University, Atlanta, Georgia, United States of America

* jente.willaert@kuleuven.be

**Data Availability Statement:** All experimental data is available on Zenodo (DOI:10.5281/zenodo.

## Abstract

Balance impairments are common in cerebral palsy. When balance is perturbed by backward support surface translations, children with cerebral palsy have increased co-activation of the plantar flexors and tibialis anterior muscle as compared to typically developing children. However, it is unclear whether increased muscle co-activation is a compensation strategy to improve balance control or is a consequence of reduced reciprocal inhibition. During translational perturbations, increased joint stiffness due to co-activation might aid balance control by resisting movement of the body with respect to the feet. In contrast, during rotational perturbations, increased joint stiffness will hinder balance control as it couples body to platform rotation. Therefore, we expect increased muscle co-activation in response to rotational perturbations if co-activation is caused by reduced reciprocal inhibition but not if it is merely a compensation strategy. We perturbed standing balance by combined backward translational and toe-up rotational perturbations in 20 children with cerebral palsy and 20 typically developing children. Perturbations induced forward followed by backward movement of the center of mass. We evaluated reactive muscle activity and the relation between center of mass movement and reactive muscle activity using a linear feedback model based on center of mass kinematics. In typically developing children, perturbations induced plantar flexor balance correcting muscle activity followed by tibialis anterior balance correcting muscle activity, which was driven by center of mass movement. In children with cerebral palsy, the switch from plantar flexor to tibialis anterior activity was less pronounced than in typically developing children due to increased muscle co-activation of the plantar flexors and tibialis anterior throughout the response. Our results thus suggest that a reduction in reciprocal inhibition causes muscle co-activation in reactive standing balance in children with cerebral palsy.

8220096). All code used for model fitting and data analysis is available on Github (https://github.com/WillaertJente/RotationalPerturbations).

**Funding:** This study was funded by the Flemish agency for scientific research (FWO-Vlaanderen) through a research fellowship to Jente Willaert (1192320N) and by the National Institute of Health (NIH) research grant NIH R01 HD46922 to Lena H. Ting. The funders had no role in study design, data collection and analysis, decision to publish, or preparation of the manuscript.

**Competing interests:** The authors have declared that no competing interests exist.

## Author summary

Children with cerebral palsy often have balance impairments. When standing balance is perturbed using support-surface translations, children with cerebral palsy have high agonist-antagonist co-activation. Increasing joint stiffness by co-activating agonists and antagonists might aid standing balance control during translational, but not rotational perturbations. If children with cerebral palsy show muscle co-activation during rotational perturbations, we can assume that this co-activation is a consequence of neurological impairments (e.g., not be able to suppress antagonist activation upon stretch of the agonist).

We found that both in typically developing children and children with cerebral palsy reactive muscle activity could be explained by delayed feedback of center of mass kinematics. Our combined translational and rotational perturbations induced a switch in center of mass movement (forward to backward). We found that upon this switch in center of mass movement, typically developing children switched from plantarflexor to tibialis anterior muscle activity. This switch in muscle activity was less clear in children with cerebral palsy, although center of mass movement was the same as in typically developing children, due to muscle co-activation. Therefore, our results suggest that neural impairments cause muscle co-activation in reactive standing balance in children with cerebral palsy.

## Introduction

Balance impairments are common in cerebral palsy (CP), but it is yet unclear which motor control pathways contribute to these balance impairments [1,2]. When balance is perturbed by backward support surface translations, children with CP have increased co-activation of the plantar flexors and tibialis anterior as compared to typically developing (TD) children [1–4]. However, it is unclear whether muscle co-activation is a compensation strategy to improve balance control or a consequence of neural deficits [5,6]. Increased joint stiffness due to muscle co-activation might aid standing balance control in response to translational perturbations by resisting movement of the body with respect to the feet. However, increased joint stiffness will hinder standing balance control in response to rotational perturbations as it induces coupling between the body and platform motion resulting in body tilt (Fig 1). Hence, if co-activation is

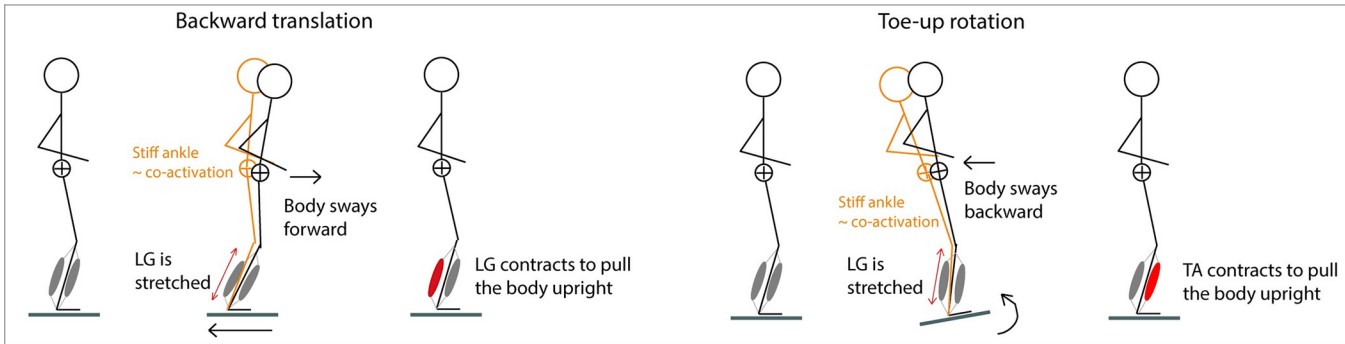

**Fig 1. Illustration of the effect of stiffening the ankle (in orange) (e.g., by co-activation) on body movement induced by a backward translation versus toe-up rotation of the platform.** A stiff ankle will help to keep the body upright in response to translational perturbations by resisting movement of the body with respect to the feet whereas it will lead to additional body tilt in response to rotational perturbations by coupling body to platform movement. Backward translational perturbations will require balance correcting activity of the plantar flexors to counteract forward body tilt whereas rotational toe-up perturbations will require balance correcting activity of the tibialis anterior to counteract backward body tilt.

used as a compensation strategy in children with CP in translational perturbations, we do not expect to observe it in response to rotational perturbations. But if co-activation is caused by neural deficits, we do expect to observe it in response to rotational perturbations as well.

Reduced reciprocal inhibition is often observed in CP [5–10] and could cause increased co-activation. Reciprocal inhibition is defined as the inhibition of antagonistic muscle activity upon agonistic muscle activation [11]. In healthy adults, rapid dorsiflexion of the ankle elicits reflex activation of the plantar flexors (stretched muscles) while the antagonistic muscle (i.e., tibialis anterior) remains silent. In contrast, in children with CP, rapid dorsiflexion of the ankle elicits a response with similar latencies in both plantar flexors and tibialis anterior. This impaired reciprocal inhibition in CP is observed with different levels of muscle activation [5,6]. When the tibialis muscle is activated to actively dorsiflex the ankle, the soleus muscle is inhibited in healthy adults [12], whereas there is no modulation of soleus inhibition during tibialis anterior contraction in children with CP [6]. Such reduction in reciprocal inhibition might also affect sensorimotor processing, (i.e., how the nervous system translates incoming sensory information about body motion into motor commands to activate muscles) underlying reactive balance control in CP.

We found that both agonists and antagonists were more sensitive to CoM disturbances elicited by support-surface translations in children with CP than in TD children [4]. Previous work in healthy and pathological humans and animals [13–17] has shown that muscle activity in response to support-surface translations can be explained by CoM kinematics, with CoM displacement, velocity, and acceleration having a distinct effect on the muscle response. In a previous study, we reconstructed plantar flexor and tibialis anterior activity elicited by support-surface translations by a weighted linear combination of delayed (to account for neural transmission times) CoM displacement, velocity, and acceleration. We included stabilizing pathways that activate the plantar flexor and tibialis anterior when the CoM moves forward and backward respectively, and a destabilizing pathway that activates the tibialis anterior when the CoM moves forward (and plantar flexor activity is required to stabilize posture) [16]. The weights or gains of the CoM displacement, velocity, and acceleration in the linear combination indicate the sensitivity of the muscle response to CoM disturbances. We found that displacement and velocity gains were higher for plantar flexors and tibialis anterior in children with CP than in TD children [4]. The higher gains for the tibialis anterior for the destabilizing pathway reflect increased co-activation and might reflect reduced reciprocal inhibition, i.e., a similar sensory input (CoM kinematics) drives both the agonist and antagonist. Yet, to distinguish reduced reciprocal inhibition from the use of co-activation as a successful compensation strategy, we should also investigate perturbations that contain a rotational component, where muscle co-activation will not help balance recovery (Fig 1).

In healthy humans and animals, CoM movement seems to dictate muscle responses to a wide range of translational and rotational support-surface perturbations. Translational and rotational perturbations that elicit similar CoM movements yet different joint movements induce activity in similar groups of muscles [17–20]. Furthermore, delayed feedback from CoM kinematics but not joint angles could explain reactive muscle activity during a train of translational perturbations that induced uncorrelated CoM and joint angle trajectories [14]. By combining backward translational and toe-up rotational perturbations, we can induce a change in the direction of the CoM displacement during the response (forward in response to translation followed by backward in response to rotation). Based on the previously proposed CoM feedback theory, a switch from plantar flexor to tibialis anterior muscle activity upon reversal of the CoM is expected in healthy subjects. However, if reciprocal inhibition is impaired in children with CP, we expect the antagonist to be activated along with the agonist hindering this switch from plantar flexor to tibialis anterior muscle activity.

Here, we combined translational and rotational perturbations to test whether muscle co-activation is the result of compensation or due to neural deficits. If increased co-activation in response to translational perturbations is due to neural deficits, we expect to see it in response to combined translational and rotational perturbations as well. In that case, the increased co-activation might be the result of reduced reciprocal inhibition leading to simultaneous activation of agonists and antagonists. First—to assess whether co-activation is due to neural deficits and not a compensation strategy—we evaluated reactive muscle activity. We hypothesized that children with CP would have prolonged activity in the plantar flexors, i.e., we expected that the plantar flexors would remain active upon reversal of the CoM movement in children with CP but not in TD children. In addition, we hypothesized that children with CP would have increased muscle co-activation as compared to TD children. Second -to assess whether plantar flexor and tibialis anterior activity might have been driven by a common input due to reduced reciprocal inhibition—we evaluated the relation between CoM movement and reactive muscle activity using the sensorimotor response model described by Welch & Ting [13]. The sensori-motor model used for this analysis describes both a stabilizing and destabilizing pathway based on CoM movement and is therefore suitable to distinguish a muscle's role as an agonist and antagonist. We hypothesized that CoM feedback could explain muscle activity in both TD children and children with CP, but that gains for the stabilizing and destabilizing pathway would be higher in children with CP.

Overall, we found that activation of the plantar flexors was prolonged leading to a less pronounced switch from plantar flexor to tibialis anterior activity upon reversal of the CoM movement in children with CP. We observed increased muscle co-activation throughout the response and increased stabilizing and destabilizing gains in children with CP. Similar as in TD children, reactive muscle activity in children with CP could be explained by delayed CoM feedback. Our results thus suggest that muscle co-activation in CP is due to neural impairments rather than being a compensation strategy during balance perturbations.

## Materials & methods

### Ethics statement

The study was approved by the Ethical Committee of UZ/KU Leuven (S63321). Forty-six children participated in this study. A legal representative of the participant and participants signed a written informed consent or informed assent form, respectively, before the start of the measurements according to the principles of the Declaration of Helsinki.

### Participants

Children with CP were recruited through the CP reference center at the University Hospital Leuven (Belgium). All patients were diagnosed as having spastic CP by a neuro-pediatrician and met the following inclusion criteria: (1) 5 to 17 years old; (2) Gross Motor Function Classification Scale (GMFCS) I-III; (3) able to stand independently for at least 10 minutes; (4) no orthopedic or neurological surgery in the previous year; and (5) no botulinum neurotoxin injections in the previous 6 months. TD children were recruited through colleagues and friends and were age matched with the children with CP.

Data from six children were excluded due to (1) the child not completing the whole protocol (N = 1), (2) the child being unable to follow the instructions (N = 3), or (3) missing EMG data due to technical errors (N = 2). Data from 20 TD children (8 girls/12 boys) and 20 children with CP (9 girls/11 boys) were included for further analysis (Table 1). Fourteen children with CP were unilaterally involved and six children with CP were bilaterally involved. Fifteen children had GMFCS level I and five children had GMFCS level II. Sixteen children had

**Table 1. Demographic data of participants (mean and standard deviations).**

| | | CP | | TD | |
|---|---|---|---|---|---|
| | | **Mean** | **(±SD)** | **Mean** | **(±SD)** |
| Female/Male | | 9/11 | | 8/12 | |
| Age | (years) | 12.3 | 3.1 | 12.0 | 3.0 |
| Length | (cm) | 153 | 16 | 156 | 16 |
| Weight | (kg) | 46.7 | 16.7 | 43.8 | 12.8 |

CP = cerebral palsy; TD = typically developing.

spasticity in the gastrocnemius muscle, as indicated by a Modified Ashworth Scale between 1 and 3 (more information in Table A in S1 Text). All children were able to stand and walk without walking aids. For the children with CP, the most affected leg, based on clinical spasticity scores, was used for further analysis, while for TD children, one leg was randomly selected.

## Materials

Trajectories of reflective skin markers (for details on marker placement, see Fig A in S1 Text) were captured by 7 infrared Vicon cameras (Vicon, Oxford Metrics, United Kingdom, 100 Hz). Activity of gastrocnemius lateralis (LG), gastrocnemius medialis (MG), soleus (SOL), and tibialis anterior (TA) was measured simultaneously through surface electromyography (sEMG, ZeroWire EMG Aurion, Cometa, Italy, 1000 Hz). Silver-chloride, pre-gelled bipolar electrodes (Ambu Blue Sensor, Ballerup, Denmark) were placed according to SENIAM guidelines [21]. Reactive balance was tested using combined translational and rotational perturbations on an instrumented, movable platform (Caren platform, Motek, The Netherlands) (Fig 2A). Children were secured using a safety harness, connected to an overhead rail to prevent falling.

## Protocol

For every participant, we collected age and anthropometrics (length and weight). We performed a clinical exam to determine the range of motion and Modified Ashworth Score (for the gastrocnemii and soleus) for the children with CP. During balance assessment, participants stood barefoot on the platform. Before the start of the assessments, participants were instructed to stand upright and maintain balance without taking a step, unless necessary to avoid falling. When participants needed to take a step, we asked them to return their feet to the starting position, which was marked with tape on the platform. Arm movement was unconstrained. The protocol consisted of four increasingly difficult perturbation levels (increased platform displacement, velocity and/or acceleration, Fig 2B). Within each perturbation level, the same eight perturbations were administered with 12s between perturbations. When the participant stepped in more than 3 out of 8 perturbations, we did not continue to the next level. If needed, rest was given between perturbation levels. The measurements described above were part of a larger single-session protocol more broadly assessing spasticity and balance.

## Data processing and analysis

Marker trajectories were processed using OpenSim 3.3 [22,23]. A generic full-body musculoskeletal model (Gait2392 with arms, [24]) was scaled based on anatomic marker positions. Joint angles were computed using OpenSim's Inverse Kinematics tool. CoM position was computed from low-pass filtered (6Hz) joint angles using OpenSim's Body Kinematic tool. Anterior-posterior CoM displacement was expressed relative to the ankle and numerically

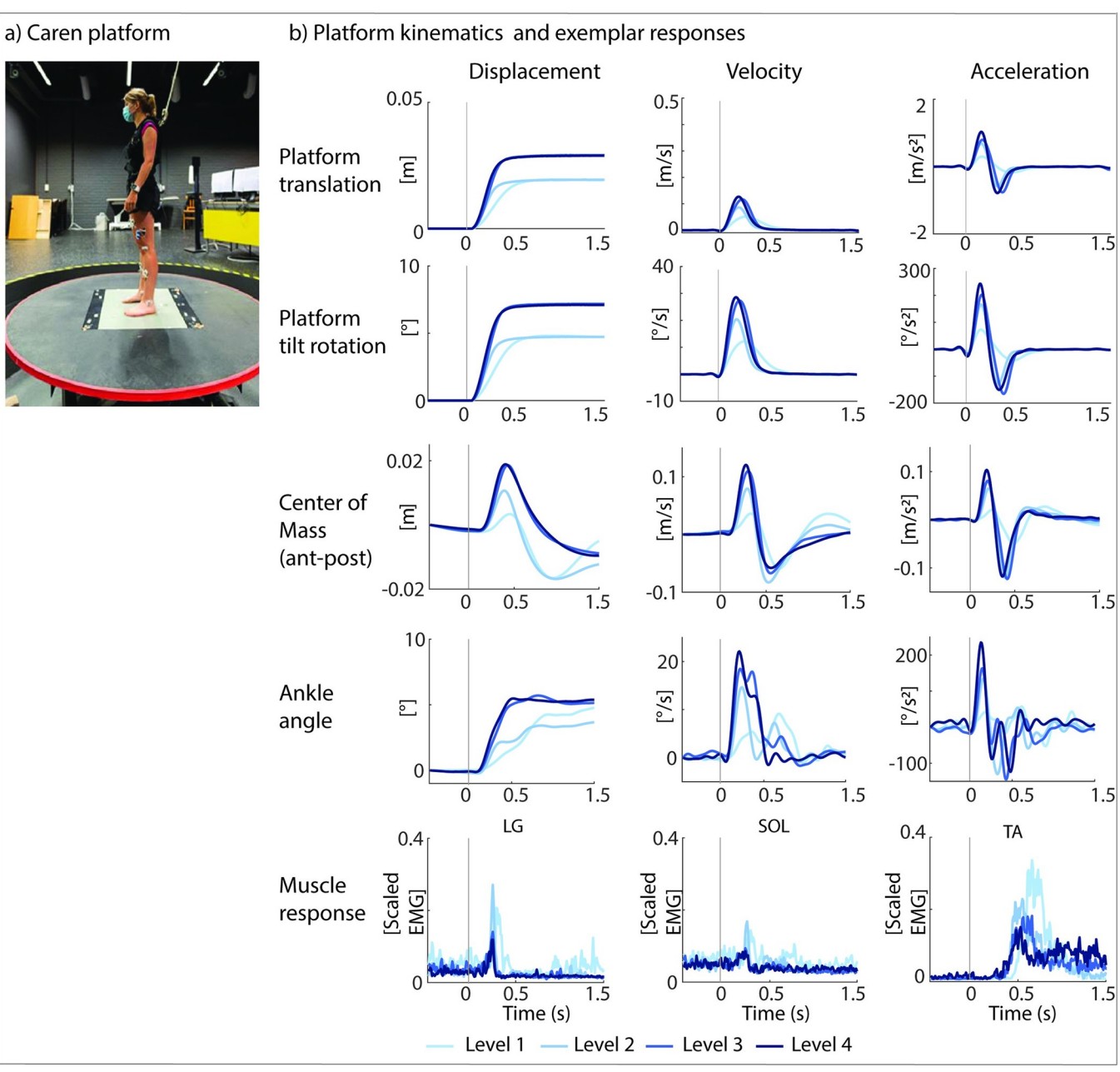

**Fig 2. Experimental set-up and perturbation profiles.** a) Caren platform for combined translational and rotational perturbations and b) platform kinematics and exemplar reactive balance responses for a typically developing child for the different perturbation levels (L1-L4). One perturbation involved simultaneous translation and rotation of the platform. Platform displacement increased from L2 to L3, while platform velocity and acceleration increased with every level. Row 1: Platform kinematics: displacement, velocity, and acceleration of a point on the platform between the ankles; row 2: Angular displacement, velocity, and acceleration; row 3: center of mass kinematics (anterior-posterior direction) relative to the ankle; row 4: ankle kinematics; row 5: muscle responses. Vertical grey line indicates platform onset LG = lateral gastrocnemius; SOL = soleus; TA = tibialis anterior.

differentiated to compute the CoM velocity. Ankle angle was numerically differentiated to compute ankle velocity. Ankle and CoM acceleration were then computed using a Savitzky-Golay filter (5$^{th}$ order and frame length 11) [25]. Average ankle angles and CoM displacement, velocity, and acceleration were calculated across all non-stepping trials within each perturbation level for each participant and each level.

EMG data was band-pass filtered using a fourth order Butterworth filter between 10 and 450 Hz followed by signal rectification. Finally, a fourth order Butterworth low-pass filter with 40 Hz cut off was applied [26]. The filtered EMG signal was scaled by the maximum, filtered value across all perturbations for every participant. As part of the bigger protocol, we also performed translational perturbations [4], which were also taken into account when determining the scaling value. The average EMG signal was calculated across all non-stepping trials for each participant and each level.

## Outcome parameters

**Muscle activity, center of mass movement, and ankle kinematics.** We computed average reactive muscle activity for LG, MG, SOL, and TA in three time bins. Reactive muscle activity was computed by subtracting baseline activity, i.e., average muscle activity in the 100ms preceding perturbation onset, from the filtered and scaled EMG. Time bins were defined based on the sequence of muscle responses during combined translational and rotational perturbations described in literature for healthy participants [20,27]. The first time bin (Z1) started at platform onset and ended 150ms later, during this time interval we observed little reactive muscle activity, possibly due to the slow movement onset in combination with neural transmission delay. The second time bin (Z2) lasted from 150ms to 250ms after platform onset, when the first peak in plantar flexor activity occurred in TD children (LG, MG, SOL). The third time bin (Z3) lasted from 250ms to 400ms after platform onset and started when a switch from plantar flexor to TA activity was typically observed in TD children (Fig 3: row 7–9).

We similarly assessed the average CoM (Fig 3: row 1–3) and ankle angle (Fig 3: row 4–6) displacement, velocity, and acceleration. All time bins were shifted 100ms backwards in time (i.e., Z1: onset platform to 50ms; Z2: 50ms to 150ms; Z3: 150ms to 300ms) as we assumed that the kinematic disturbances provided sensory inputs for the later muscle responses. Assessing the kinematics allowed us to identify potential kinematic triggers for the switch from plantar flexor to TA activity.

We assessed differences in average reactive muscle activity and kinematics for the three different time bins between children with CP (Fig 3 - left) and TD children (Fig 3- right).

**Co-contraction index.** We calculated the co-contraction index (CCI) as the overlap in filtered and scaled EMG between TA and respectively LG, MG, and SOL (PF) [28]:

$$CCI = \frac{\sum_{1}^{end} (\min(EMG_{PF}, EMG_{TA}))}{\#\text{frames}}$$

Low CCI values indicate that both muscles are active at different moments whereas high CCI values indicate that both muscles are active at the same time. We performed the analysis for two time intervals, i.e. between perturbation onset and 400ms after perturbation onset (interval covering the time bins described above), and between 0.5s before perturbation onset and 1.5s after perturbation onset (similar time interval as sensorimotor response model described below).

**Sensorimotor response model.** We tested the relation between muscle activity and CoM kinematics by reconstructing measured EMG trajectories by a weighted sum of delayed CoM acceleration, velocity, and displacement trajectories [13]. We modeled both a balance correcting and an antagonistic feedback pathway (Fig 4A) [16]. The balance correcting pathway yields plantar flexor activity when the CoM moves forward and tibialis anterior activity when the CoM moves backward. The antagonistic pathway yields plantar flexor activity when the CoM moves backward and tibialis anterior activity when the CoM moves forward. We used the following sensorimotor response model to reconstruct plantar flexor (LG, MG, SOL) activity

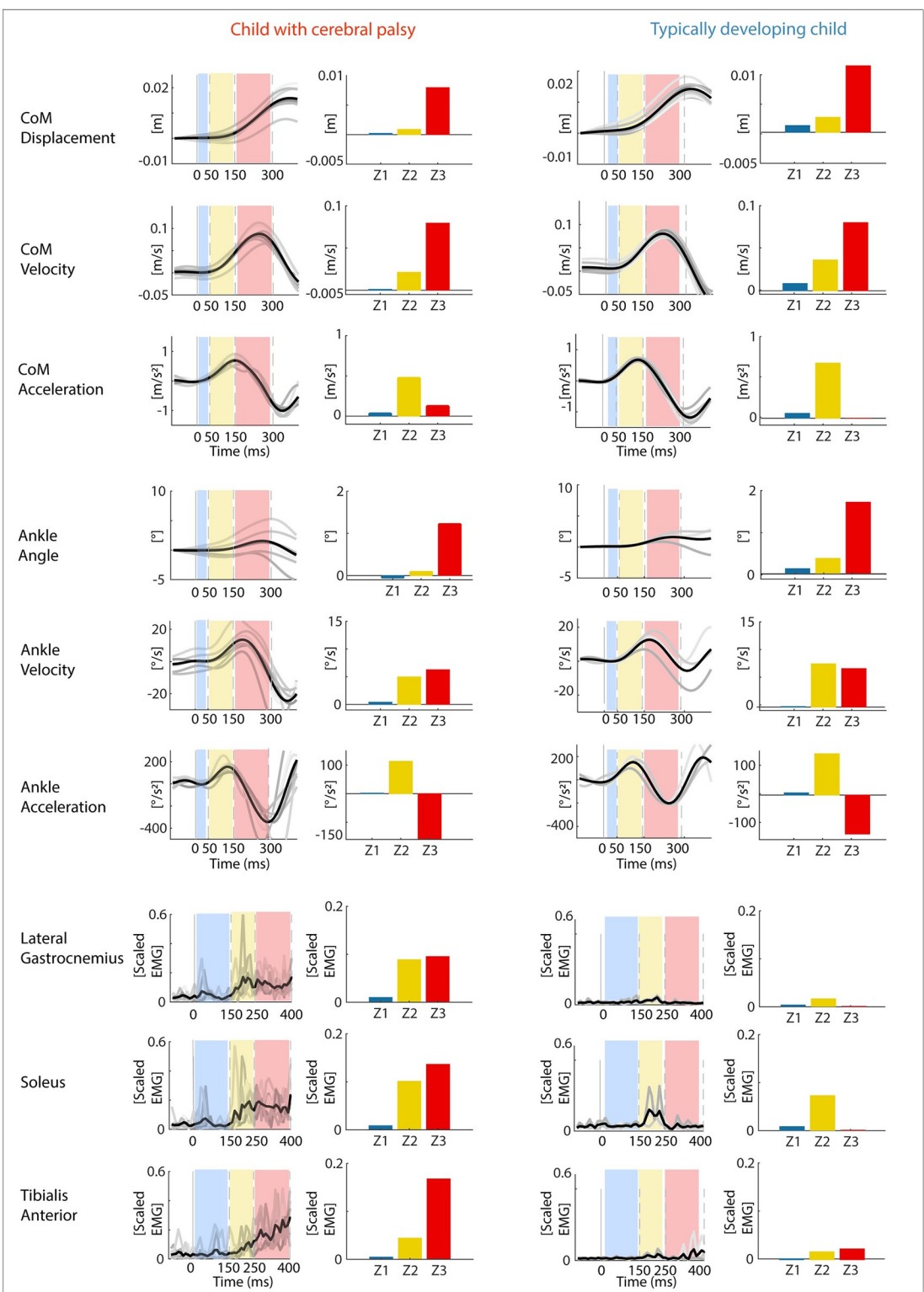

**Fig 3. Representative examples for center of mass movement, ankle kinematic, and muscle activity for perturbation level 2 in time bins (zones) for a child with cerebral palsy (left) and typically developing child (right). Both children had moderate co-activation (exemplar trajectories for children with high/low co-activation can be found in Figs B and C in S1 Text).** Row 1–3: Center of mass kinematics (displacement, velocity, and acceleration) as a function of time with indication of time bins (dotted lines, colored boxes) and average trajectories (black). Time bin 1 (Z1) in blue, time bin 2 (Z2) in yellow, and time bin 3

(Z3) in red; Row 4–6: Ankle angle kinematics (angle, velocity, and acceleration) as a function of time with indication of time bins and average trajectories. Row 7–9: Muscle activations as a function of time with indications of time bins and average muscle activity. Light gray traces are separate trials of one subject. The bars represent the average for each time bin for the corresponding (black) trace (average over trials) on the left.

(plantar flexors needed to restore upright position when CoM moves forward):

$$EMG_{recon} = \lfloor e_0 + k_d * d_{CoM}(t-\tau) + k_v * v_{CoM}(t-\tau) + k_a * a_{CoM}(t-\tau) + k_s * a_{CoMinit}(t-\tau) \rfloor \text{ (balance correcting pathway)}$$

$$+$$

$$\lfloor k_d'* - d_{CoM}(t-\tau) + k_v'* - v_{CoM}(t-\tau) + k_a'* - a_{CoM}(t-\tau) \rfloor \text{ (antagonistic pathway)}$$

and the following sensorimotor response model to reconstruct tibialis anterior activity (tibialis anterior needed to restore upright position when CoM moves backward):

$$EMG_{recon} = \lfloor e_0 + k_d* - d_{CoM}(t-\tau) + k_v* - v_{CoM}(t-\tau) + k_a* - a_{CoM}(t-\tau) \rfloor \text{ (Balance correcting pathway)}$$

$$+$$

$$\lfloor k_d'*d_{CoM}(t-\tau) + k_v'*v_{CoM}(t-\tau) + k_a'*a_{CoM}(t-\tau) \rfloor \text{ (antagonistic pathway)}$$

With $EMG_{recon}$ the reconstructed muscle activity for the plantar flexors and tibialis anterior, $e_0$ baseline muscle activity; $d_{CoM}$, $v_{CoM}$, $a_{CoM}$ center of mass displacement, velocity, and acceleration; $k_d$, $k_v$, $k_a$ feedback gains or weights for the balance correcting pathway, $k_d'$, $k_v'$, $k_d'$ feedback gains or weights for the antagonistic pathway, and a common time delay ($\tau$) of 100 ms to account for processing and neural transmission time. Only the positive part of the signal ($\lfloor . \rfloor$) was used to represent excitatory drive to motor pools. We added a separate feedback term for the initial center of mass acceleration, $a_{CoM\_init}$, with a corresponding stiction gain, $k_s$, inspired by Welch and Ting [13]. The initial burst in EMG is proportional to the initial CoM acceleration and might be driven by the initial strong increase in spindle firing that coincides with short-range stiffness in the muscle [29,30]. We therefore included this term until the change in the ankle angle was 0.5˚, corresponding to the estimated short-range stiffness range [31]. In contrast to Welch and Ting [13], we included both an acceleration and stiction term based on preliminary analyses. No stiction gain was used in the tibialis anterior muscle, as in our protocol, this muscle was first shortened and short-range stiffness is strongly reduced with prior movement [30].

Baseline activity ($e_0$) was set to the measured data 0.5s before perturbation onset. Gains were estimated by minimizing the cost function, which was defined as the weighted sum of the squared difference between reconstructed and measured EMG over a time interval from 0.5s before until 1.5s after perturbation onset (weight of 1) and squared prime gains (i.e., gains of the antagonistic pathway) (weight of 1E-4). Prime gains (i.e., gains of the antagonistic pathway) were penalized to discourage the use of the antagonistic pathway unless this had a considerable effect on the fit between reconstructed and measured EMG as we did not expect the antagonistic pathway to be active in TD children. All gains were constrained between 0 and 10/m (displacement and prime displacement), 0 and 10s/m (velocity and prime velocity), and 0 and 10s$^2$/m (acceleration, prime acceleration, and stiction). Gains were calculated for each participant and each perturbations level that was performed successfully.

We assessed the goodness of fit between measured and reconstructed muscle activity (perturbation onset until 1.5s after perturbation onset) using the coefficient of determination ($r^2$, calculated as the squared correlation coefficient), the variability accounted for (VAF, defined as the uncentered $r^2$) and root mean square error (RMSE). Furthermore, we tested for

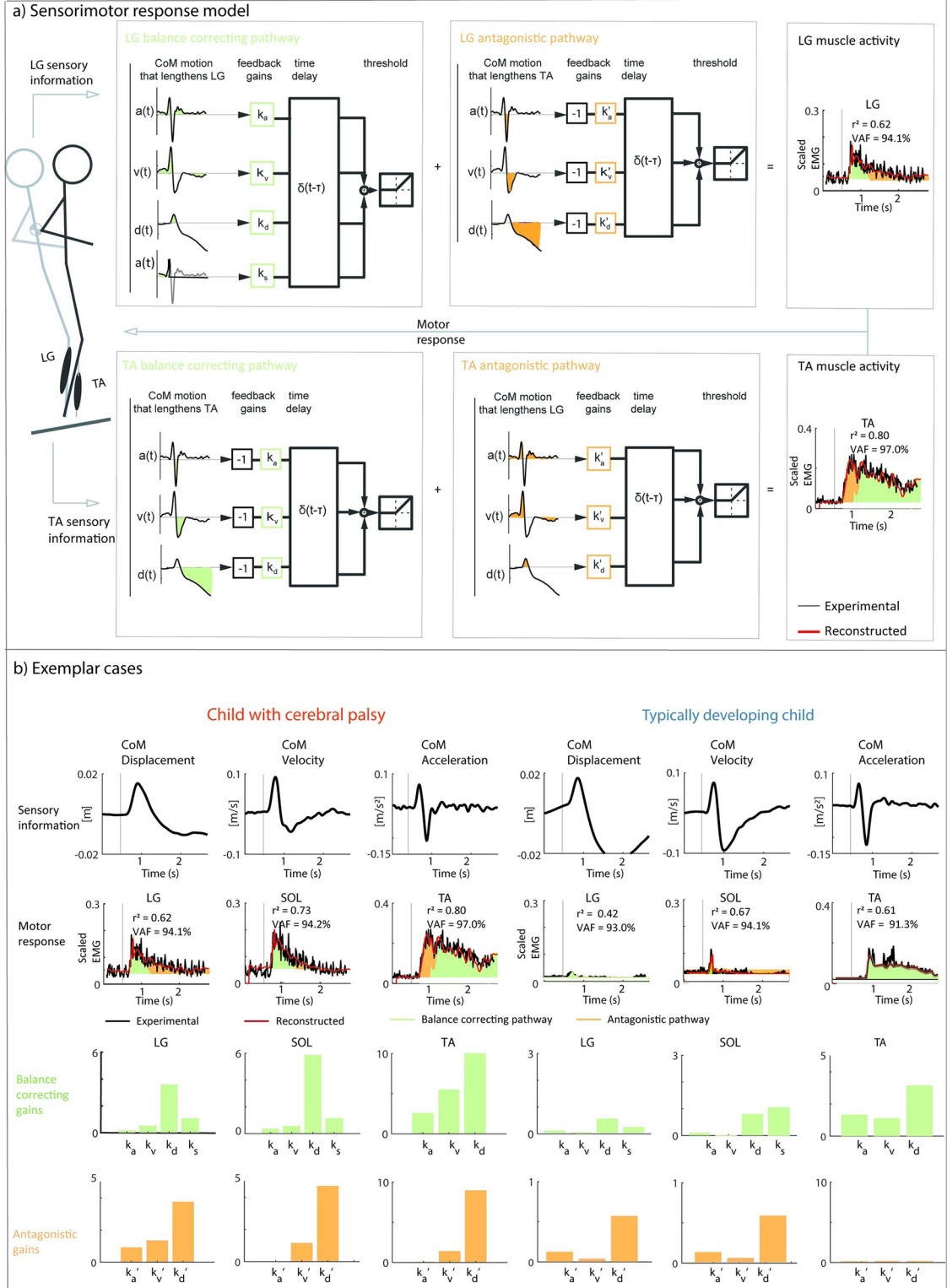

**Fig 4. Sensorimotor response model.** a) Graphical representation of the (extended) sensorimotor response model. Measured muscle activity (black) is reconstructed (red) using delayed feedback of CoM acceleration, velocity, displacement, and stiction (for plantar flexors only) along a balance-correcting pathway (green) and antagonistic pathway (orange; sensitive to CoM movement in the opposite direction). CoM acceleration, velocity, position, and stiction are multiplied by subject specific feedback gains (balance correcting: $k_a$, $k_v$, $k_d$, $k_s$; antagonistic pathway: $k_a'$, $k_v'$, $k_d'$). Note the opposite sign of the CoM kinematics in the balance correcting

(green) and antagonistic pathways (orange) of the plantar flexors (top row) and tibialis anterior (bottom row). In the extended model, both pathways are used (balance correcting and antagonistic gains), while in the simple model only the gains of the balance correcting pathway are used. b) Exemplar cases with the extended sensorimotor response model for one child with cerebral palsy (left) and one typically developing child (right) with average co-activation for perturbation level 2. Exemplar cases for children with high and low co-activation are presented in Fig E in S1 Text. Top row: center of mass kinematics; second row: measured (black) and reconstructed (red) muscle activity signals with balance correcting contribution in green and antagonistic contribution in orange; third row: balance correcting gains; bottom row: antagonistic gains (i.e., prime gains). Grey line indicated onset of perturbation. LG = lateral gastrocnemius; SOL = soleus; TA = tibialis anterior.

differences in gains between children with CP and TD children (Fig 4B). Different gains would indicate that the sensitivity to CoM perturbations differs between both groups of children.

We also explicitly tested whether EMG trajectories could be reconstructed without using the antagonistic feedback pathways (i.e., activity of the plantar flexors when the CoM moves backwards and activity of the tibialis anterior when the CoM moves forward) in both groups of children (Fig 4A). Antagonistic co-activation is not captured in the simple model. Therefore, if co-activation is present, the fit between modeled and experimental activation would be worse for the simple model compared to the extended model. We expected that the antagonistic pathway is not necessary to reconstruct reactive muscle activity in TD children, whereas adding the antagonistic pathway would improve the fit between reconstructed and measured EMG when reciprocal inhibition is impaired in children with CP. To test this, we calculated the difference in cost (squared difference between reconstructed and measured activations) between the extended model with the antagonistic pathway (characterized by prime gains) and the simple model without antagonistic pathway (no prime gains). We expressed this improvement as a percentage of the cost of the model without antagonistic pathways.

## Statistical analysis

All statistical analyses were performed using Matlab (R2018b, Mathworks, United States).

Differences in average muscle activity (for LG, MG, SOL, TA), center of mass kinematics (displacement, velocity, and acceleration) and ankle angle kinematics (change, angular velocity, and angular acceleration) between children with CP and TD children were tested for each perturbation level using a linear mixed model with two fixed effects: (1) group (CP vs. TD) and (2) time bin (1 to 3, ordinal). A participant factor was included as a random factor nested within group. We investigated whether there was a significant interaction effect between group and time bin. If this interaction effect appeared significant, post-hoc comparisons were performed.

The co-contraction index was compared between children with CP and TD children using a linear mixed model with two fixed effects: (1) group (CP vs. TD) and (2) perturbation level (1 to 4, ordinal). A participant factor was included as a random factor nested within group. We investigated whether there was a significant fixed effect of group.

Differences in gains between groups were tested using a linear mixed model with two fixed effects: (1) group (CP vs. TD) and (2) perturbation level (1 to 4, ordinal) for each muscle separately. A participant factor was included as a random factor nested within group. We investigated whether there was a significant fixed effect of group.

Improvements in cost for the extended model compared to the simple model were compared between children with CP and TD children using a linear mixed model with two fixed effects: (1) group (CP vs. TD) and (2) perturbation level (1 to 4, ordinal). A participant factor was included as a random factor nested within group. We investigated whether there was a significant fixed effect of group.

To account for unequal variances, we incorporated weights $\omega$ for each observation (i) that is used as input to the model. Weights were calculated as the inverse of the variance observed across all observations within one group (i.e., CP or TD) and condition (i.e., different time zones or perturbation level) $(y_i)$ : $\omega_i = \frac{1}{var(y_i)}$. To account for simultaneous interference, we performed the Bonferroni-Holm correction. We rapport significance before and after correction as p-values before correction give an indication of the raw associations between variables.

## Results

Due to technical errors in EMG recordings, we had to exclude data of LG in one TD child, SOL in one child with CP, and TA in another child with CP. All TD children performed all levels. Four children with CP did not perform levels 2–4 and one child with CP did not perform levels 3–4.

### Muscle activity, center of mass movement, and ankle kinematics

**Muscle activity.** Children with CP modulated their muscle response differently in time than TD children, indicating a reduced ability to switch between plantar flexors and tibialis anterior (Fig 5 and Table B in S1 Text, significant interaction effect between time bin and group). Muscle activity changed differently from the second time bin to the third time bin in children with CP compared to TD children (except for L1 in MG and L4 in SOL) (Tables C and D in S1 Text). The decrease in plantar flexor activity (LG, MG, SOL) from the second (increasing acceleration of CoM) to the third time bin (decreasing acceleration of CoM, i.e., forward movement of CoM slows down) that was present in TD children was reduced or even absent in children with CP (Tables C and D in S1 Text). The increase in tibialis anterior muscle activity from the second to the third time bin was higher in children with CP than in TD children (Tables C and D in S1 Text).

**Center of mass movement.** CoM displacement, velocity, and acceleration were different between time bins but not between groups, indicating that there were no differences in CoM movement between groups (Fig 6A and Table E in S1 Text). While average CoM displacement and velocity continued to increase from the second to the third time bin, average CoM acceleration decreased from the second to third time bin. Hence, CoM acceleration is a potential trigger for the switch between plantar flexor and TA activity.

**Ankle kinematics.** Ankle displacement, velocity, and acceleration were different between time bins but not between groups (Fig 6B and Table F in S1 Text). There was an interaction effect between time bins and groups for angular velocity (L1-L3) (Tables G and H in S1 Text). Angular velocity increased less from time bin 2 to 3 in children with CP than in TD children (Fig 6B). While average ankle angular displacement and velocity continued to increase from the second to the third time bin, average ankle angular acceleration decreased from the second to third time bin (Fig 6B). Hence, ankle angular acceleration is another potential trigger for the switch between plantar flexor and tibialis anterior activity.

### Co-contraction index

Co-activation was higher in children with CP than in TD children over the entire response. The CCI was significantly higher for children with CP than for TD children across all perturbation levels and for all muscle pairs (LG and TA, MG and TA, SOL and TA) for both the shorter (0-400ms after perturbation onset) (Fig 7 and Tables I and J in S1 Text) and longer (0.5s before until 1.5s after perturbation onset) time interval (Tables K and L and Fig D in S1 Text).

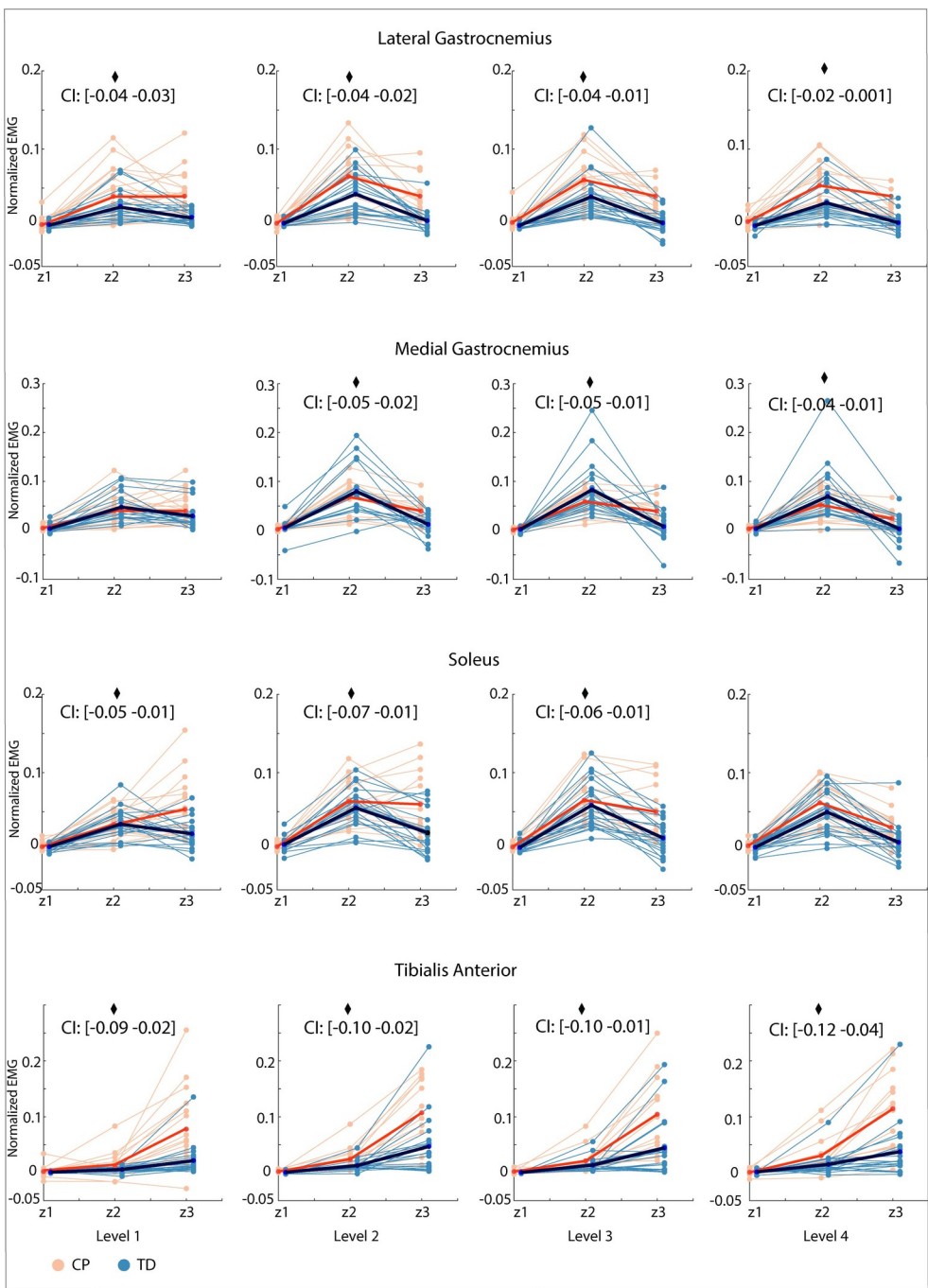

**Fig 5. Average normalized EMG for three time bins (Z1-Z3) for all muscles and all levels.** Children with cerebral palsy (CP) in orange, typically developing (TD) children in blue. Group average for children with cerebral palsy in red, group average for typically developing children in dark blue. Significant interaction effects after Bonferroni-Holm correction between time bin and group (p < 0.05) are indicated with a diamond.

## Sensorimotor response model

Muscle EMG during combined translational and rotational perturbations of standing can be reconstructed by delayed CoM feedback in both children with CP and TD children as reflected in the high goodness of fit values ($r^2$—CP: $0.60 \pm 0.21$, TD $0.50 \pm 0.20$; VAF–CP: 90.04% $\pm$

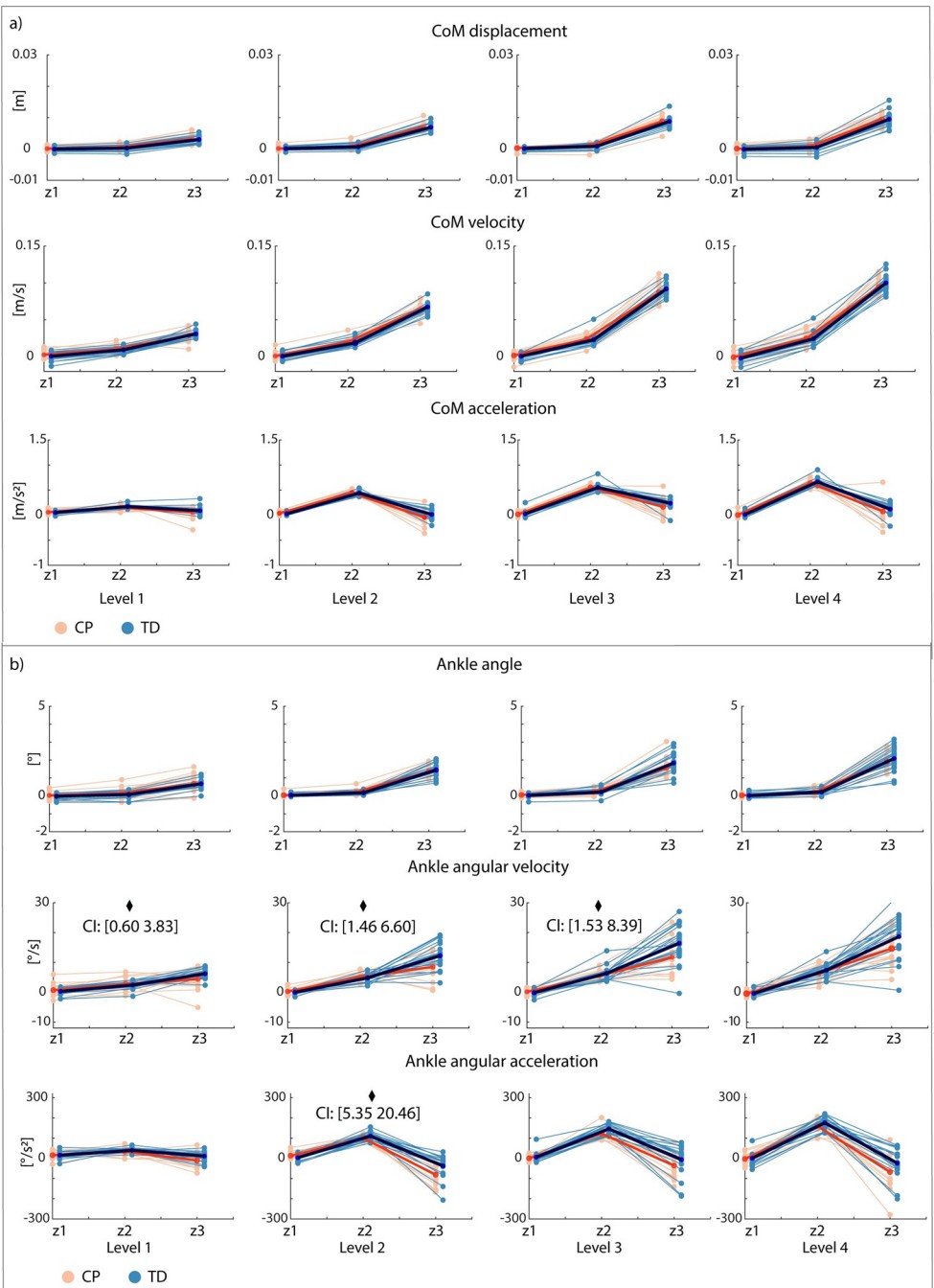

**Fig 6. Average center of mass (a) and ankle kinematics (b) for three time bins (Z1-Z3) for all levels.** Children with cerebral palsy (CP) in orange, typically developing (TD) children in blue. Group average for children with cerebral palsy in red, group average for typically developing children in dark blue. Significant interaction effects after Bonferroni-Holm correction between time bin and group (p < 0.05) are indicated with a diamond.

6.9%, TD: 86.9% ± 7.1%) and low error scores (RMSE–CP: 0.018 ± 0.01, TD: 0.015 ± 0.01) (Fig 8A and Table M in S1 Text).

The fitting error (defined as the squared error between measured and reconstructed signals) was lower when adding the antagonistic pathways for both children with CP (LG: 26.4% ±

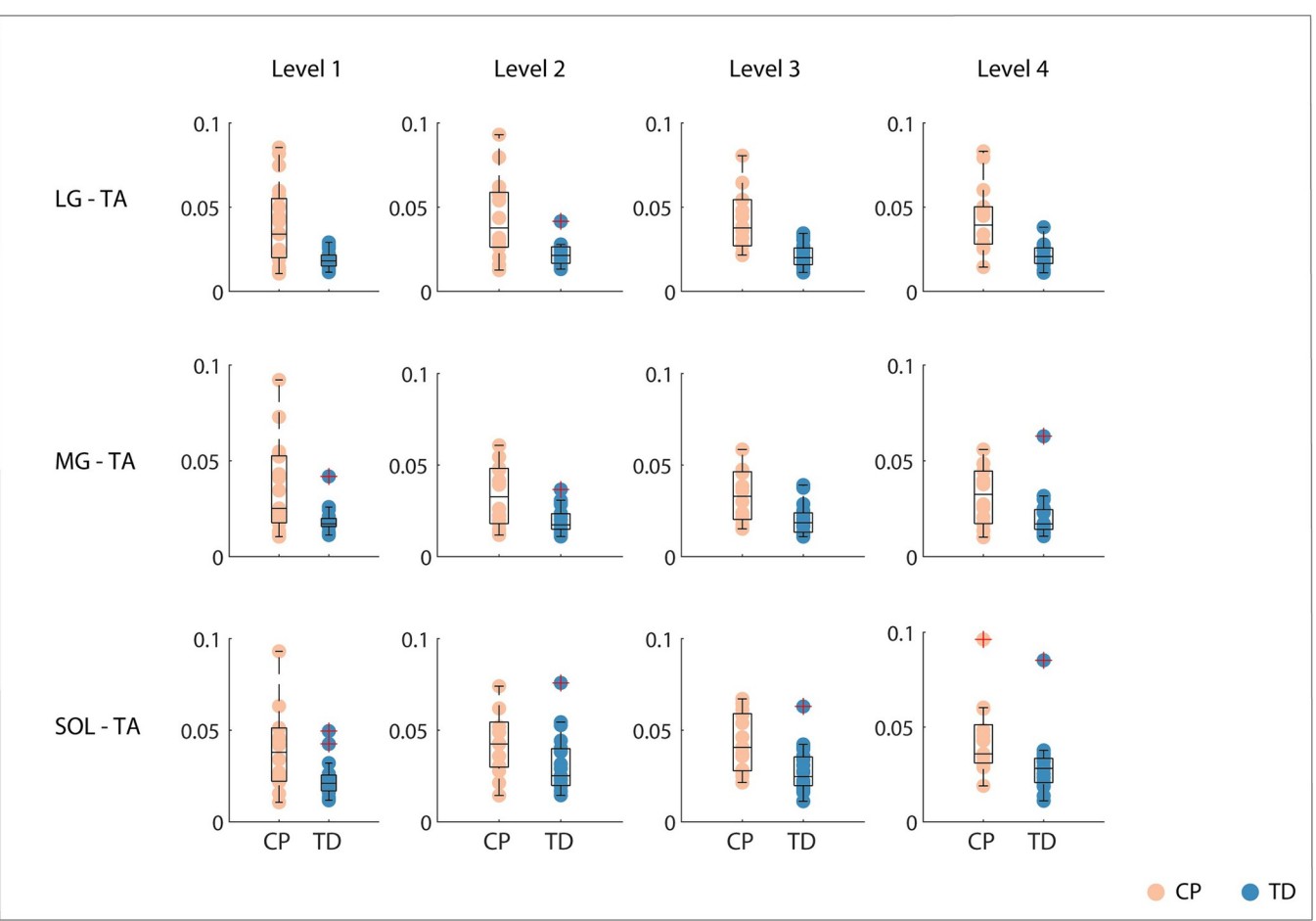

**Fig 7. Co-contraction index for perturbation onset until 400ms after perturbation onset.** Children with cerebral palsy (CP) in orange, typically developing (TD) children in blue. Boxplots in black indicate median and interquartile range and dots represent individual scores. Groups are significantly different across all levels for all muscle pairs. LG = lateral gastrocnemius; MG = medial gastrocnemius; SOL = soleus; TA = tibialis anterior.

3.7%; MG: 23.4% ± 4.5%; SOL: 26.0% ± 8.5%; TA: 63.2% ± 6.8%) and TD children (LG: 8.2% ± 4.5%; MG: 5.5% ± 2.6%; SOL: 12.1% ± 3.8%; TA:43.9% ± 5.1%) (Fig 8B and Table N in S1 Text). Adding the antagonistic pathway induced a larger reduction in the fitting error for children with CP than for TD children for the soleus (p = 0.010, CI:[-31.73–4.46]) and tibialis anterior (p = 0.041, CI:[-27.806–0.616]) across all levels.

Children with CP had higher gains for CoM feedback than TD children for all muscles (Fig 9 and Tables O and P and fig F in S1 Text).

For the lateral gastrocnemius, velocity (average increase of 203% across all levels, p = 0.004, CI: [-0.915–0.183]), displacement (91%, p = 0.018 (not significant after Bonferroni-Holm correction), CI: [-2.937–0.275]), and prime velocity (728%, p = 0.003, CI: [-0.804–0.173]) gains were higher in children with CP than in TD children across all levels (Fig 9A and Table P in S1 Text).

For the medial gastrocnemius, velocity (69%, p = 0.037 (not significant after Bonferroni-Holm correction), CI:[-0.754–0.024]), prime acceleration (281%, p = 0.014 (not significant after Bonferroni-Holm correction), CI:[-0.856–0.097]), prime velocity (973%, p < 0.001, CI: [-0.835–0.406]), and prime displacement (292%, p = 0.048 (not significant after Bonferroni-Holm correction), CI:[-2.256–0.013] gains were higher in children with CP than in TD children across all levels (Fig F and Table P in S1 Text).

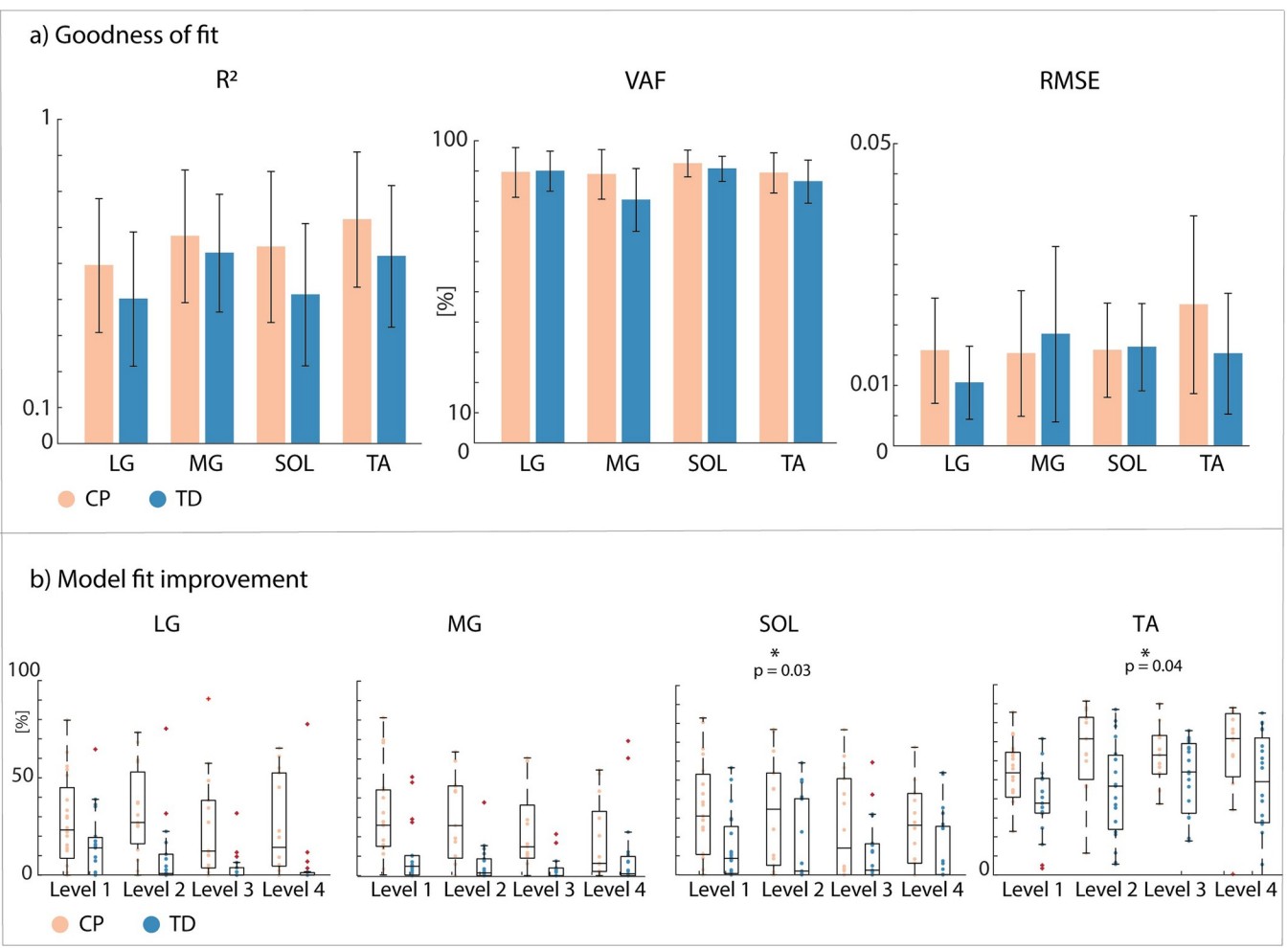

**Fig 8. Goodness of fit for the extended sensorimotor response model and reduction in cost for the extended versus simple sensorimotor response model.** a) Goodness of fit values and error scores across all levels for all muscles for the extended (with antagonistic pathways) sensorimotor response model. Standard deviations are indicated in black. $r^2$ = r squared; VAF = variance accounted for; RMSE = root mean square error. b) Improvement in cost (%) when using the extended versus simple (no antagonistic pathways) sensorimotor response model for all muscles. The cost is the squared difference between measured and reconstructed EMG. Boxplots in black indicate median and interquartile range, dots represent individual scores. Significant differences between groups (CP vs. TD) are indicated with a star and p-values. LG = lateral gastrocnemius; MG = medial gastrocnemius; SOL = soleus; TA = tibialis anterior. Children with cerebral palsy (CP) in orange, typically developing (TD) children in blue.

For the soleus, velocity (112%, p < 0.001, CI:[-0.810–0.234]), displacement (25%, p = 0.026 (not significant after Bonferroni-Holm correction), CI:[-2.469–0.185]), prime acceleration (141%, p = 0.001, CI:[-0.946–0.238]), and prime velocity (197%, p = 0.010 (not significant after Bonferroni-Holm correction), CI:[-0.970–0.133]) gains were higher in children with CP than in TD children across all levels (Fig F and Table P in S1 Text).

For the tibialis anterior, velocity (p = 0.002, CI:[-2.671–0.650]), prime velocity (p < 0.001, CI:[-1.774–0.488]), and prime displacement (p = 0.036 (not significant after Bonferroni-Holm correction), CI:[-4.227–0.148]) gains were higher in children with CP than in TD children across all levels (Fig 9B and Table P in S1 Text).

## Discussion

Reduced reciprocal inhibition might contribute to balance impairments in CP. We perturbed standing balance in children with CP and TD children by a combined translational and

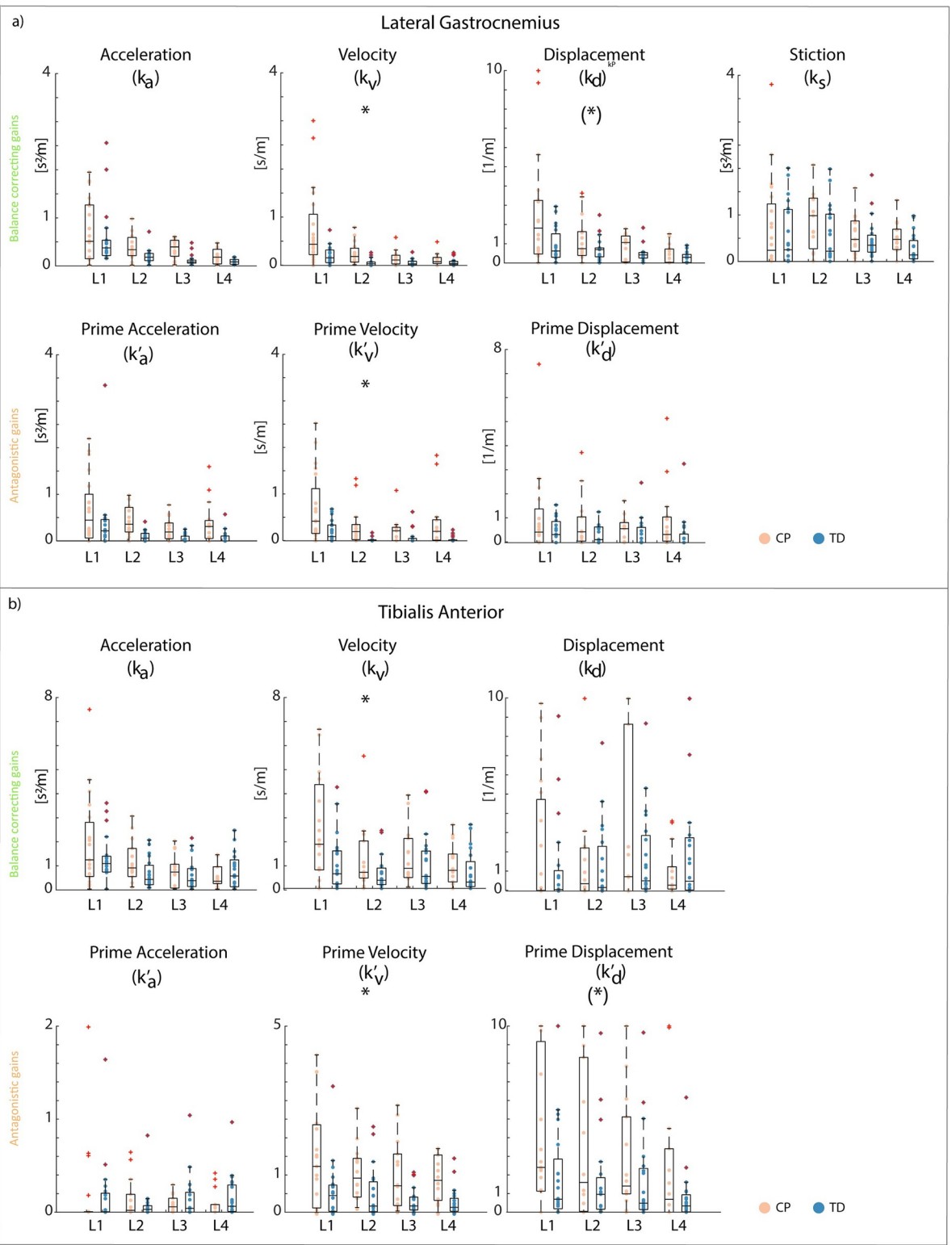

**Fig 9. Center of mass feedback gains of children with cerebral palsy and typically developing children. a) Lateral gastrocnemius. b) Tibialis Anterior.** Differences between children with CP and TD children were larger for lateral gastrocnemius than for soleus and medial gastrocnemius, therefore results for lateral gastrocnemius are shown in the main figure. Figures for medial gastrocnemius and soleus can be found in Fig F in S1 Text. Upper row: balance correcting pathway gains, bottom row: antagonistic pathway gains. L1-L4: Level 1 to level 4. Boxplots in black indicate mean and interquartile ranges and dots represent individual scores. Children with cerebral palsy (CP) in orange,

typically developing (TD) children in blue. Significant differences between groups are indicated with a star after Bonferroni-Holm correction. A star between brackets indicates significant effects that did not survive the Bonferroni-Holm correction.

rotational movement of the support surface that induced an initial plantar flexor response, followed by a switch to tibialis anterior activity in TD children. This switch from plantar flexor to tibialis anterior activity was less pronounced in children with CP (Fig 5). The reduced ability to selectively activate muscles in response to CoM movement was reflected in increased co-activation of plantar flexors and tibialis anterior throughout the response (Fig 7) and in the gains of the sensorimotor response model (Fig 9). In both children with CP and TD children, muscle activity in response to combined translational and rotational perturbations could be explained by delayed CoM feedback. However, we found that accounting for antagonistic pathways was more needed in children with CP than in TD children to reconstruct measured EMG (Fig 8). In addition, feedback gains of both the balance-correcting and antagonistic pathways were higher in children with CP than in TD children meaning that muscle activity was more sensitive to CoM disturbances in children with CP than in TD children. Higher joint stiffness, as induced by muscle co-activation, hinders balance control in response to rotational perturbations where retaining an upright posture depends on the ability to dissociate the motion of the body from the motion of the feet. Our results thus suggest that muscle co-activation in CP is due to neural impairments rather than being a compensation strategy during balance perturbations. We did not explore models based on feedback from joint instead of CoM kinematics because previous work has demonstrated that models based on feedback from joint kinematics could not explain muscle responses across perturbation conditions [14]. However, CoM and ankle kinematics might be related in some parts of the response (see e.g., Fig 6) and hence the switch from plantar flexor to dorsiflexor activity also coincides with a change in sign of ankle angular acceleration.

Decreased reciprocal inhibition is a common symptom in children with CP, contributing to excessive antagonistic muscle co-activation during voluntary movement such as walking [5,6,9]. Although increased muscle co-activation was observed in response to translational perturbations of standing balance in CP, this was not tested before during combined translational and rotational perturbations. Our results show that when tibialis anterior was activated, the plantar flexors were silenced in TD children but remained active in children with CP, suggesting that the expected reciprocal inhibition of the plantar flexors is lacking in children with CP [5].

Our observation that delayed CoM feedback could explain muscle activity in response to combined rotational and translational perturbations suggests that task-level feedback is preserved in children with CP. We previously demonstrated that CoM feedback could explain muscle activity in response to support surface translations in children with CP [4]. Since support surface translations elicit correlated CoM and ankle movements, we could not confidently conclude that task-level feedback was preserved in CP. Backward translational perturbations cause forward CoM movement whereas toe-up rotational perturbations cause backward CoM movement. Both perturbations dorsiflex the ankle (note that the observed body movement is the result of both the platform movement and the response of the subject). Using such combined translational and rotational perturbations, it was demonstrated that muscle activity is dictated by CoM movement in healthy adults [19,20,27]. Healthy adults first activate their gastrocnemius and then switch to activating their tibialis anterior [20,27]. Although these experiments demonstrated that reactive muscle activity is driven by CoM movement rather than by joint movement during combined translational and rotational perturbations, we tested this for the first time–to our knowledge–using an explicit sensorimotor response model in both TD children and children with CP.

We are unable to attribute alterations in responsive muscle activity to local or task-level feedback pathways. Both spinal and supraspinal feedback pathways are involved in reactive balance [32] and both pathways are impaired in CP. Children with CP often have reflex hyper-excitability reflecting spinal feedback deficits [7,33]. In addition, they have difficulties with selectively controlling their muscles due to impaired supraspinal pathways [7,33]. Reflex hyper-excitability might lead to exaggerated responses of both the plantar flexors, which are stretched during balance perturbations, and the antagonistic tibialis anterior. Such increased sensitivity to muscle stretch might prevent children with CP to suppress plantar flexor activity, requiring compensations in supraspinal control to stabilize posture. Alternatively, the reduced ability to selectively activate muscles might rely in the supraspinal pathways. To gain more insight in the underlying pathways, we explored the relation between the gains and a clinical score of joint hyper-resistance, i.e., the MAS score of the gastrocnemius (exploratory analysis, Figs G and H in S1 Text). We did not find any correlations. However, this might be due to inherent limitations of the MAS score [34]. For example, the MAS score does not distinguish contributions from reflex hyper-excitability and altered passive tissue properties to joint hyper-resistance. A more in-depth analysis is thus required.

Notwithstanding the altered control of reactive balance, CoM displacements were not larger in children with CP than in TD children. We found that the maximum CoM excursion in response to translational perturbations was smaller in children with CP than in TD children [4]. This might indicate that co-activation indeed helps to maintain an upright position when standing is perturbed by translational perturbations (although it might have causes children with CP to step at lower perturbation levels). Here, we found no differences in maximum CoM excursion between both groups. This suggests that children with CP were able to compensate for the increased co-activation when controlling the CoM position in response to rotational perturbations. We observed differences in the ankle angle velocity and acceleration, suggesting less ankle dorsiflexion movement. Children with CP thus likely compensated in other joints to control their CoM position.

Fewer children with CP than TD children were able to successfully perform the higher perturbation levels, leading to missing data in our analysis. It is likely that the children who were unable to perform the higher perturbation levels were more affected and therefore differed more form TD children. Given that we found differences between groups notwithstanding missing data from more affected children only strengthens our conclusions. The maximum EMG across all perturbation levels was used for scaling and therefore missing data might have impacted EMG scaling. However, it is unlikely that EMG magnitude at a given perturbation level is similar across children given the differences in muscle responses that we observed. Therefore, our scaling method is based on the assumption that the EMG magnitude is similar at maximal performance, i.e., at the highest perturbation level that could be performed without stepping. Many TD children might have been able to withstand perturbations larger than those admitted here without stepping. Hence, we might have used scaling factors that are too low, resulting in an overestimation of the muscle responses and thus an underestimation of the differences between groups.

Our results might have clinical importance and give inspiration to therapist to explore new training options. Currently, clinical assessments mainly focus on hyper-reflexia, whereas reduced reciprocal inhibition is evaluated less. Based on our results, we suggest that clinical assessments of reduced reciprocal inhibition might provide useful information about balance control impairments. Therapists should be aware of the underlying mechanisms of muscle co-activation during balance control in order to develop novel therapeutic approaches. These new therapies should specifically target reciprocal inhibition as our results suggest that improved reciprocal inhibition might improve balance control in children with CP.

## Conclusion

Reduced reciprocal inhibition might underlie altered standing balance control in children with CP, leading to increased co-activation in response to both support surface translations and rotations. Notwithstanding this reduced reciprocal inhibition, task-level control of the CoM was preserved in the rather functional (GMFCS I and II) children with CP included in this study. Future work should explore whether alterations in balance control are also correlated with falls.

## Supporting information

**S1 Text. Including: S1. Additional information on children with cerebral palsy. Table A. Additional information on children with cerebral palsy**. GMFCS = Gross Motor Function Classification System (range 1–5) H = Hemiplegic; D = Diplegic; L = Left; R = Right; MAS = Modified Ashworth Scale for the gastrocnemii (range 1–4). **S2. Full body marker set-up. Fig A. Marker set-up. S3. Muscle activity, center of mass movement, and ankle kinematics**. **S3.1 Exemplar trajectories**. **Fig B. Exemplar trajectories for center of mass movement, ankle kinematics, and muscle activity for perturbation level 2 in time bins (zones) for a child with cerebral palsy (left) and typically developing child (right) with low co-activation. FigC. Exemplar trajectories for center of mass movement, ankle kinematics, and muscle activity for perturbation level 2 in time bins (zones) for a child with cerebral palsy (left) and typically developing child (right) with high co-activation. S3.2 Muscle activity**. **Table B. Statistical outcome parameters (p-values) for EMG time bins for children with cerebral palsy and typically developing children.** LG = lateral gastrocnemius; MG = medial gastrocnemius; SOL = soleus; TA = tibialis anterior. Significant differences ($p < 0.05$) are indicated in bold. **Table C. Post-hoc comparison for the interaction effect between group and time bin for EMG time bins for children with cerebral palsy and typically developing children.** LG = Lateral Gastrocnemius; MG = Medial Gastrocnemius; SOL = Soleus; TA = Tibialis Anterior. Significant differences are indicated in bold (before Bonferroni-Holm correction). **Table D. Statistical outcome parameters (p-values, Fstat, and confidence intervals) for the interaction effect between time bin and group.** LG = lateral gastrocnemius; MG = medial gastrocnemius; SOL = soleus; TA = tibialis anterior; BH = Bonferroni-Holm.Significant results are indicated with Y (yes) in column six before Bonferroni-Holm correction and in column nine after Bonferroni-Holm correction. New alpha-levels defined by the Bonferroni-Holm correction are indicated in column eight. **S3.3 Center of mass movement**. **Table E. Statistical outcome parameters (p-values) for CoM time bins for children with cerebral palsy and typically developing children.** Significant differences are indicated in bold. **Table F. Statistical outcome parameters (p-values) for ankle angle kinematics for children with cerebral palsy and typically developing children.** Significant differences are indicated in bold. **Table G. Post-hoc comparison for the interaction effect between group and time bin for angular velocity and acceleration for children with cerebral palsy and typically developing children.** LG = Lateral Gastrocnemius; MG = Medial Gastrocnemius; SOL = Soleus; TA = Tibialis Anterior. Significant differences are indicated in bold (before Bonferroni-Holm correction). **Table H. Statistical outcome parameters (p-values, Fstat, and confidence intervals) for the interaction effect between time bin 3 and group.** Pos = Ankle angle position; Vel = Ankle angle velocity; Acc = Ankle angle acceleration; BH = Bonferroni-Holm. Significant results are indicated with Y (yes) in column six before Bonferroni-Holm correction and in column nine after Bonferroni-Holm correction. New alpha-levels defined by the Bonferroni-Holm correction are indicated in column eight. **S4. Co-contraction index**. **Table I. Statistical outcome parameters (p-values) for co-contraction index between children with cerebral palsy and**

**typically developing children for the time frames similar as the time bins (onset-350ms after onset).** LG = lateral gastrocnemius; MG = medial gastrocnemius; SOL = soleus; TA = tibialis anterior. Significant differences are indicated in bold. **Table J. Statistical outcome parameters (p-values, Fstat, and confidence intervals) for the fixed effect of group.** LG = lateral gastrocnemius; MG = medial gastrocnemius; SOL = soleus; TA = tibialis anterior. BH = Bonferroni-Holm. Significant results are indicated with Y (yes) in column six before Bonferroni-Holm correction and in column nine after Bonferroni-Holm correction. New alpha-levels defined by the Bonferroni-Holm correction are indicated in column eight.

**Table K. Statistical outcome parameters (p-values) for co-contraction index between children with cerebral palsy and typically developing children for the time frames similar as the sensorimotor response model.** LG = lateral gastrocnemius; MG = medial gastrocnemius; SOL = soleus; TA = tibialis anterior. Significant differences are indicated in bold. **Table L. Statistical outcome parameters (p-values, Fstat, and confidence intervals) for the fixed effect of group.** LG = lateral gastrocnemius; MG = medial gastrocnemius; SOL = soleus; TA = tibialis anterior. BH = Bonferroni-Holm. Significant results are indicated with Y (yes) in column six before Bonferroni-Holm correction and in column nine after Bonferroni-Holm correction. New alpha-levels defined by the Bonferroni-Holm correction are indicated in column eight. **Fig D. Co-contraction index for 0.5s before perturbation onset until 1.5s after perturbation onset.** Children with cerebral palsy (CP) in orange, typically developing (TD) children in blue. Grey bars indicate group averages, boxplots in black indicate median and interquartile range and dots represent individual scores. Groups are significantly different across all levels for all muscle pairs. LG = lateral gastrocnemius; MG = medial gastrocnemius; SOL = soleus; TA = tibialis anterior. **S5. Sensorimotor response model**. **S5.1 Exemplar responses**. **Fig E. Exemplar trajectories for the sensorimotor response model for children with low co-activation (a) and children with high co-activation (b) S5.2 Goodness of fit values**. **Table M. Goodness of fit and error scores (mean and standard deviations) for the extended sensorimotor response model for children with cerebral palsy and typically developing children.** $R^2$: R-squared, indicating fit with overall pattern; VAF = variance accounted for, indicating fit with amplitude of response activity; RMSE = root mean square error, indicating absolute error between measured and reconstructed signal. LG = lateral gastrocnemius; MG = medial gastrocnemius; SOL = soleus; TA = tibialis anterior. CP = cerebral palsy; TD = typically developing. **S5.3 Improvement in fit when adding antagonistic muscle pathways**. **Table N. Improvement in fit (mean and standard deviations) when adding antagonistic feedback pathways (extended model vs. simple model) for children with cerebral palsy and typically developing children.** LG = lateral gastrocnemius; MG = medial gastrocnemius; SOL = soleus; TA = tibialis anterior. CP = cerebral palsy; TD = typically developing. Significant differences (p<0.05) are indicated in bold. **S5.4 Feedback gains**. **Table O. Statistical outcome parameters (p-values) for feedback gains for the extended model for children with cerebral palsy and typically developing children.** LG = lateral gastrocnemius; MG = medial gastrocnemius; SOL = soleus; TA = tibialis anterior. ka = acceleration gain; kv = velocity gain; kd = displacement gain; ka' = prime acceleration gain; kv' = prime velocity gain; kd' = prime displacement gain; ks = stiction gain. Significant differences are indicated in bold. **Table P. Statistical outcome parameters (p-values, F-stat, and confidence intervals) for the fixed effect of group.** LG = lateral gastrocnemius; MG = medial gastrocnemius; SOL = soleus; TA = tibialis anterior. BH = Bonferroni-Holm. Significant results are indicated with Y (yes) in column six before Bonferroni-Holm correction and in column nine after Bonferroni-Holm correction. New alpha-levels defined by the Bonferroni-Holm correction are indicated in column eight. **Fig F. Center of mass feedback gains for all levels for children with cerebral palsy and typically developing children.** a) Medial

gastrocnemius. b) Soleus. Upper row: balance correcting pathway gains, bottom row: antagonistic pathway gains. L1-L4: level 1 to level 4. Boxplots in black indicate mean and interquartile ranges, and dots represent individual scores. Children with cerebral palsy (CP) in orange, typically developing (TD) children in blue. Significant differences between groups are indicated with a star after Bonferroni-Holm correction. A star between brackets indicates significant effects that did not survive the Bonferroni-Holm correction. **S6. Correlation with MAS. Fig G. Associations between feedback gains for the lateral gastrocnemius, medial gastrocnemius and soleus and the Modified Ashworth Score of the gastrocnemii.** Dots are individual scores. Gains for lateral gastrocnemius in light orange, medial gastrocnemius in orange, soleus in dark orange. **Fig H. Associations between feedback gains for the tibialis anterior and the Modified Ashworth Score of the gastrocnemii.** Dots are individual scores.
(DOCX)

## Acknowledgments

We would like to thank all our participants for participating in this study.

## Author Contributions

**Conceptualization:** Lena H. Ting, Friedl De Groote.

**Data curation:** Jente Willaert.

**Formal analysis:** Jente Willaert, Friedl De Groote.

**Funding acquisition:** Jente Willaert, Kaat Desloovere, Lena H. Ting, Friedl De Groote.

**Investigation:** Jente Willaert, Friedl De Groote.

**Methodology:** Jente Willaert, Lena H. Ting, Friedl De Groote.

**Project administration:** Jente Willaert.

**Resources:** Kaat Desloovere, Anja Van Campenhout, Friedl De Groote.

**Software:** Jente Willaert.

**Supervision:** Kaat Desloovere, Anja Van Campenhout, Lena H. Ting, Friedl De Groote.

**Validation:** Jente Willaert, Friedl De Groote.

**Visualization:** Jente Willaert, Friedl De Groote.

**Writing – original draft:** Jente Willaert.

**Writing – review & editing:** Jente Willaert, Kaat Desloovere, Anja Van Campenhout, Lena H. Ting, Friedl De Groote.

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
