## [Decision Letter · Decision Letter 0]

16 Jan 2024

Dear Ms. Willaert,

Thank you very much for submitting your manuscript "Combined translational and rotational perturbations of standing balance reveal contributions of reduced reciprocal inhibition to balance impairments in children with cerebral palsy" for consideration at PLOS Computational Biology.

As with all papers reviewed by the journal, your manuscript was reviewed by members of the editorial board and by several independent reviewers. In light of the reviews (below this email), we would like to invite the resubmission of a significantly-revised version that fully takes into account the reviewers' comments.

We cannot make any decision about publication until we have seen the revised manuscript and your response to the reviewers' comments. Your revised manuscript is also likely to be sent to reviewers for further evaluation.

Sincerely,

Lizeth Sloot, Ph.D.

Guest Editor

PLOS Computational Biology

Lyle Graham

Section Editor

PLOS Computational Biology

Reviewer's Responses to Questions

**Comments to the Authors:**

Reviewer #1: Please see attachment

Reviewer #2: The proposed manuscript is a combined experimental and computational modeling study of the potential mechanisms of muscle co-contraction in children with cerebral palsy (CP). It uses translational and rotational perturbations to evaluate whether co-contraction of the TA and soleus are pathological consequences of CP or a motor control strategy. A computational model is then employed to test the hypothesis that co-contraction is caused by decreased reciprocal inhibition observed in children with CP. Th experimental design is quite clever and the use of modeling aids in overcoming some of the experimental limitations for examining mechanisms of CP symptoms. However, the study rationale is somewhat lost in the presentation of the introduction and there are some methodological choices that must be better justified. Finally, the statistical approach is incomplete and detracts from interpreting the data faithfully.

Major Issues

Introduction

- One of my primary concerns about the paper is that there is, in fact, two (admittedly related and interesting) studies here but they are being interwoven into a more elaborate story that makes the rationale of the study a bit convoluted. From what I can tell, there are two relevant observations from previous studies that motivate this one: 1) children with CP have increased muscle co-contraction and 2) children with CP have decreased reciprocal inhibition. The current study hypothesizes that 1) increased co-contraction is pathological, i.e. not a control strategy AND 2) that the increased co-contraction is caused by decreased reciprocal inhibition. The experimental paradigm tests the first and the modeling work tests the second. It is a reasonable rationale and approach, but it took me several reads to actually extract those couple sentences. (I actually just deleted an entire paragraph in this feedback critiquing the rationale of this study being fundamentally flawed because I misunderstood the rationale as it is currently presented.) I would recommend a rewrite that lays out the study rationale more linearly to avoid these confusions. I think it is critical to state that the experimental data tests only whether increased co-contraction is pathological vs a control strategy (there is no testing reciprocal inhibition at this point). It establishes that the co-contraction is not a control strategy and the subsequent modeling work implicates reciprocal inhibition.

Methods

- The sensorimotor response model should be better justified. Why is a relatively simple weighted sum of delayed COM acceleration, velocity, and displacement used to model EMG? What physiological, e.g. neuronal or musculoskeletal, properties or phenomenon motivate this model? Is the simple addition of a negative sign appropriate to create an ‘antagonist pathway’? Is it reasonable that COM be used to describe EMG for muscles which only span a limited number of joints?

- There is a severe overreliance on p-values in this paper and it results in a statistical approach that, while complex and extensive, undermines the results and the reader’s confidence in them. For starters, the statistical section does not seem to indicate the use of any corrections for multiple tests, which is very concerning. But even with these corrections, the huge number of inference tests being performed make it very difficult to be confident in each result. Ideally, I want to recommend that the authors remove the extraneous tests, focusing instead on the a priori hypotheses of the study and use more descriptive statistical metrics (confidence intervals, effect sizes, etc.) where appropriate. The purpose of hypothesis tests is just that – hypothesis testing.

o One example of how this approach could be problematic is in Figure 6b. The manuscript states:

“Ankle angular velocity and acceleration were different between time bins and there was an interaction effect between time bin and group for most perturbation levels (velocity: L1, L2, L3, p < 0.02; acceleration: L1, L2, p < 0.05)….Visual inspection reveals a lower angular velocity and larger angular deceleration in the last time bin in children with CP”

This text states that there is a difference between time bins and then suggests that while there is no overall difference between the two groups, there is an interaction, i.e. the combination of group and time bin, which is significant. But we’ve already established that there are differences between time bins so having “some” of the interaction terms come up as significant really doesn’t indicate any type of difference between the groups. It is inappropriate to imply that “there are some differences between the groups in some time bins” based on the statistical testing used here. With this in mind, the statement “Visual inspection reveals a lower angular velocity and larger angular deceleration in the last time bin in children with CP” seems somewhat misleading. The decreased angular velocity is maybe a couple of degrees per second? Is that significant or clinically relevant, particularly with a limited sample number? Is the quality of the OpenSim InverseKinematics simulations so robust that this difference is above possible noise? It seems like if confidence intervals were used, they would likely be nearly identical. Can we really say these are different? I focused on Fig 6, but it applies to Fig 5 as well.

Forgive me for pontificating on this point so fervently, but I feel it is important. I would suggest that the authors revisit their statistical tests and report not only the p-value but the effect sizes and confidence intervals as well and use the entirety of information to inform their conclusions.

Results

- In Figure 5 and 6, it looks as though average values are displayed for each time bin, but these are actively modulating signals (EMG and kinematics). Is it appropriate to simply average across this whole time period?

Discussion

- I think it would be worthwhile for the authors to elaborate on the implications of their findings, particularly for treatment of CP. Any speculation should be explicitly stated as such, but it would be good to provide a little more context to the paper’s findings.

Minor Issues

- How are the time bins (Z1, Z2, and Z3) selected? Based on physiology?

- Figure 3 & 4: While I appreciate the authors’ transparency in labelling their example as ‘exemplar’ but it immediately begs the question, “So if these are the ‘best’, what do the typical examples look like?” I admit there are already a lot of plots in these figures so maybe including other examples in the supplementary would be useful. One approach is to include three examples: best, representative, and worst.

- Figure 3: I’m a bit confused what the bar plots actually show? Are they the mean of each time bin for the corresponding traces to the left? It would be good to include this information in the legend.

**Have the authors made all data and (if applicable) computational code underlying the findings in their manuscript fully available?**

Reviewer #1: None

Reviewer #2: Yes

PLOS authors have the option to publish the peer review history of their article (what does this mean?). If published, this will include your full peer review and any attached files.

Reviewer #1: No

Reviewer #2: No
---

## [Decision Letter · Decision Letter 1]

25 Apr 2024

Dear Ms. Willaert,

Thank you very much for submitting your manuscript "Combined translational and rotational perturbations of standing balance reveal contributions of reduced reciprocal inhibition to balance impairments in children with cerebral palsy" for consideration at PLOS Computational Biology. As with all papers reviewed by the journal, your manuscript was reviewed by members of the editorial board and by several independent reviewers. The reviewers appreciated the attention to an important topic. Based on the reviews, we are likely to accept this manuscript for publication, providing that you modify the manuscript according to the review recommendations.

Sincerely,

Lizeth Sloot, Ph.D.

Guest Editor

PLOS Computational Biology

Lyle Graham

Section Editor

PLOS Computational Biology

Reviewer's Responses to Questions

**Comments to the Authors:**

Reviewer #1: Please note that the sentence numbers refer to the document with tracked changes.

Overall, the previous comments have been addressed well, and I like the paper and its contribution. I just have two general comments where I think some extra explanation is required, and some smaller comments about writing/grammar.

General comments:

Line 425-426: Is the EMG normalization affected by the fact that not all patients were able to complete the protocol? If the perturbations are not as large for some participants as for others, while the maximum across all is used for normalization, it seems possible to me that this normalization could affect the results. Has this been investigated?

Line 580-581: here, only the COM is written as a potential trigger, while the results have shown that also the ankle angular acceleration is a potential trigger. Please motivate why only the COM is listed in the discussion.

Writing comments:

Line 93-98: this sentence is complicated. I would suggest splitting it into two.

Line 142-148: consider rephrasing as the sentence is quite complex.

Line 172: Please rename section to Participants

Line 208: connected instead of adjusted?

Figure 2 caption, 2nd line: exemplar reactive balance responses

Line 247-248: was this maximum value before or after filtering?

Line 333-334: I would put this information closer to the equation as it is quite relevant for the equation.

Line 416-419: I did not understand this part. Consider rephrasing.

Line 451 and 459: were instead of was

Figure 7: what does the light grey mean in the figure?

Line 649: evaluated less (word order)

Reviewer #2: The authors have appropriately addressed my previous feedback and concerns. The introduction more clearly articulates the study rationale and hypotheses as well as the previous work that formulated the study. Furthermore, I applaud the changes to the statistical approach which is much more comprehensive and, in my opinion, adds greater confidence to the findings.

**Have the authors made all data and (if applicable) computational code underlying the findings in their manuscript fully available?**

Reviewer #1: Yes

Reviewer #2: Yes

PLOS authors have the option to publish the peer review history of their article (what does this mean?). If published, this will include your full peer review and any attached files.

Reviewer #1: No

Reviewer #2: No

Figure Files:

Data Requirements:

Reproducibility:

References:

---

## [Editor Report · Decision Letter 2]

28 May 2024

Dear Ms. Willaert,

We are pleased to inform you that your manuscript 'Combined translational and rotational perturbations of standing balance reveal contributions of reduced reciprocal inhibition to balance impairments in children with cerebral palsy' has been provisionally accepted for publication in PLOS Computational Biology.

Best regards,

Lizeth Sloot, Ph.D.

Guest Editor

PLOS Computational Biology

Lyle Graham

Section Editor

PLOS Computational Biology

---

## [Editor Report · Acceptance letter]

5 Jun 2024

PCOMPBIOL-D-23-01258R2 

Combined translational and rotational perturbations of standing balance reveal contributions of reduced reciprocal inhibition to balance impairments in children with cerebral palsy

Dear Dr Willaert,

I am pleased to inform you that your manuscript has been formally accepted for publication in PLOS Computational Biology. Your manuscript is now with our production department and you will be notified of the publication date in due course.

With kind regards,

Lilla Horvath
